# When Do Causal Fairness Constraints Work? Reproducing and Stress-Testing Long-Term Fair Reinforcement Learning

## Abstract

We study the reproducibility of *A Causal Lens for Learning Long-Term Fair Policies* by Lear & Zhang (2025), which introduces *qualification gain disparity* (QGD) as a long-term fairness objective in sequential decision-making and proposes causality-aware PPO variants (PPO-C and PPO-Cb) to reduce it. Building on the authors' official implementation, we replicate their core experiments in a bank-lending task and test whether the reported disparity reductions, causal decomposition trends, and utility–fairness trade-offs hold. Our results largely confirm the original findings: PPO-C and PPO-Cb consistently reduce QGD relative to standard PPO and fairness-aware baselines, with the causal decomposition suggesting that these reductions mainly come from making the learned policy's direct treatment of groups more similar, rather than from changes in the environment's transition dynamics. However, we find that utility preservation is weaker than originally reported in some settings. We further extend the evaluation along three axes: strongly imbalanced population ratios, a $K$-group extension (where $K > 2$) based on Qualification Gain Variance (QGV), and a structurally different infectious-disease environment. These extensions show that the $K$-group objective is highly sensitive to the fairness coefficient: untuned penalties can collapse utility, while moderate values recover useful trade-offs. We also show that group-level causal decomposition remains diagnostically useful, with reductions in QGV arising mainly through the direct policy component while structural sources of disparity are offset by indirect dynamics rather than eliminated. Overall, we support most of the original claims while clarifying when causal long-term fairness objectives remain effective and stable.

## 1 Introduction

As machine learning and artificial intelligence increasingly participate in decision-making across high-risk areas such as education, mortgage lending, criminal justice risk assessment, and emergency triage (Baker & Hawn, 2022; Lee & Floridi, 2021; Berk et al., 2021; Sánchez-Salmerón et al., 2022), biases inherent in data, such as those related to gender, ethnicity, and race, may be inadvertently amplified by algorithms, leading to discriminatory or unfair outcomes. Consequently, balancing decision fairness with model performance within machine learning systems has become a matter of growing concern. However, as numerous studies have shown, most fairness research focus solely on immediate bias within static scenarios, overlooking the potential long-term consequences of decisions within dynamic systems (D'Amour et al., 2020; Nashed et al., 2023). This narrow emphasis on short-term fairness may fail to safeguard the long-term interests of vulnerable groups.

To address this, some researchers have proposed modelling the delayed impact of decisions and individual reactions in order to measure the deferred effects of decisions on fairness (Liu et al., 2020). In particular, Yin et al. (2023), Ge et al. (2021) and Hu et al. (2023b) have studied fairness in reinforcement learning (RL) settings. RL provides a natural framework for sequential decisions where early actions affect later states, allowing bias to accumulate over time and enabling the study of long-term fairness beyond one-shot scenarios. However, optimizing reward alone doesn't ensure long-term fairness and may conflict with short-term fairness constraints (e.g., making policies blind to sensitive attributes can hinder long-term equity, Hu et al., 2024). To address this, Lear & Zhang (2025) proposed a causally-based, long-term fairness decomposition framework.

Unlike short-term fairness metrics such as demographic parity or equal opportunity, this approach focuses on fairness issues that emerge gradually over time as agents make dynamic decisions. Furthermore, the authors employ causal decomposition to break down long-term fairness differences into direct, indirect, and environmentally spurious effects, thereby examining how long-term unfairness can still be present despite short-term constraints being satisfied.

## 2 Scope of Reproducibility

Lear & Zhang (2025) propose *qualification gain parity* (QGD) for a long-term fairness objective in sequential decision-making: different sensitive groups should, over time, accumulate similar increases in *qualification*, not just receive similar immediate outcomes. They measure unfairness as the gap in expected cumulative qualification gain and decompose it into *Direct Policy Effect* (DPE), *Indirect Policy Effect* (IPE), and *Spurious Policy Effect* (SPE), further explained in section 3.4. They then introduce two algorithms based on proximal policy optimization (PPO, Schulman et al., 2017), namely, PPO-C, which penalizes qualification gain disparity directly, and PPO-Cb, which additionally regularizes for short-term benefit fairness.

The focus of our work is on reproducing their following core claims:

1. The proposed algorithms, PPO-C and PPO-Cb, achieve lower QGD than standard PPO and existing fairness-aware baselines.
2. Reducing QGD via PPO-C and PPO-Cb incurs only a limited loss in utility relative to PPO and remains competitive with existing fairness-aware baselines.
3. PPO-Cb improves immediate benefit fairness and reduces the DPE relative to PPO-C.
4. The causal decomposition of QGD is a useful diagnostic, showing that the strongest disparity reductions for PPO-C and PPO-Cb come mainly from decreasing the DPE, the direct effect.

## 3 Methodology

Our methodology is organized around two objectives: reproducing the original binary-group lending experiments of Lear & Zhang (2025) and stress-testing the proposed causal fairness objectives beyond that setting. For the reproduction, we follow the released implementation as closely as possible, preserving the environment dynamics, PPO-based training procedure, model architecture, and hyperparameter choices. For the extensions, we modify the evaluation setting along three axes: population imbalance, $K$-group fairness (where $K > 2$), and transfer to an infectious-disease control environment. The section proceeds as follows: Section 3.1.1 describes the baseline and causality-aware PPO variants; Section 3.2 details the lending environment and implementation; the following subsections define the training protocol and evaluation metrics; and Sections 3.5–3.6 describe the infectious-disease and $K$-group extensions.

### 3.1 Model descriptions

#### 3.1.1 Baselines

We compare against three established algorithms, following the released implementation of Lear & Zhang (2025). **PPO** (Schulman et al., 2017) is the unconstrained utility-maximization baseline; following the original work, we use the KL-penalized objective rather than the clipped surrogate. **A-PPO** (Yu et al., 2022) regularizes the advantage function, subtracting a penalty when the current state violates a fairness metric and penalizing actions predicted to worsen it. **F-PPO-L** (Hu et al., 2023a) augments the PPO objective with a long-term fairness regularizer based on the 1-Wasserstein distance between the groups' outcome distributions, weighted by a parameter $\lambda$, omitting any short-term constraints.

#### 3.1.2 Causality-Based PPO

PPO-C (Lear & Zhang, 2025) integrates a causal fairness constraint into PPO's objective, specifically, with an additional term based on *qualification gain disparity* between advantaged and disadvantaged groups Let $C_\pi(\theta)$ denote this disparity under the current policy $\pi_\theta$, i.e., the difference in expected cumulative

qualification gain between groups induced by $\pi_\theta$. This long-term fairness metric is decomposed into three causal components: DPE reflecting direct instantaneous bias, IPE capturing long-term accumulated disparity indirect of a policy's immediate actions, and SPE for remaining disparity that is not due to policy decisions but rather due to external or inherent differences in the environment. Section 3.4 goes into more detail on these three components. Although they are analysed separately, PPO-C's training objective initially incorporates the overall parity constraint as a squared penalty term added to the PPO objective, with a weight $\beta^C$.

$$J(\theta) = L^{UTIL} - \beta^{KL}L^{KL} - \beta^C(\hat{C}_{\pi_\theta})^2. \tag{1}$$

Here, $L^{\text{UTIL}}(\theta)$ denotes the standard PPO surrogate objective, corresponding to expected advantage term with importance sampling, and $L^{\text{KL}}(\theta)$ is a KL-divergence regularizer that discourages large updates from the previous policy. PPO-C therefore differs from the baseline only through the additional QGD penalty term.

### 3.1.3 PPO-C with Benefit Fairness

PPO-Cb (Lear & Zhang, 2025) extends PPO-C by adding a second fairness regularizer to the objective (in addition to the qualification parity constraint $C_\pi$): a second fairness penalty (denoted $\Lambda$), inspired by the notion of *individual benefit fairness* (Plecko & Bareinboim, 2023), measuring whether individuals who would benefit equally from a positive decision are treated similarly. This term is formulated analogously as a Gini-style coefficient measure over cross-group benefit differences, and PPO-Cb incorporates it via an additional penalty weighted by $\beta^\Lambda$.

$$J(\theta) = L^{UTIL} - \beta^{KL}L^{KL} - \beta^C(\hat{C}_{\pi_\theta})^2 - \beta^\Lambda\Lambda. \tag{2}$$

Intuitively, PPO-Cb places extra emphasis on long-term outcome equity: it looks to equalize not only the overall qualification gains (like PPO-C does) but also to ensure that the policy does not preferentially allocate opportunities to one group when members of both groups stand to gain similarly. By tuning the weight on this benefit fairness term, PPO-Cb prioritizes reducing delayed effects of bias even if it allows a slight increase in short-term inequality.

## 3.2 Experimental Setup and Code

Lear & Zhang (2025) provide a Python implementation of the lending environment and RL algorithms (j-proj, 2025) used in their paper. The codebase is built on OpenAI Gym for simulation and Stable-Baselines3 (PyTorch) for PPO training. This provided a clear starting point for this reproducibility study[1]. The original configuration settings were used as a reference to reproduce the main experimental settings, including for the environment dynamics and most hyperparameters.

### 3.2.1 Environment and Setup

We restate the sequential lending environment introduced by Lear & Zhang (2025). At each time step $t$, the agent (lender) observes an applicant characterized by a discrete credit score $x_t$ and a sensitive attribute $s \in \{s^+, s^-\}$. The agent takes a binary decision $d_t \in \{0, 1\}$ (deny/approve). This decision influences the applicant's future credit score: approved loans result in score updates based on repayment or default, while denied loans are subject only to exogenous drift. The agent's objective is to maximize expected cumulative reward (profit).

We evaluate the three environment variants defined by the original study, which differ in their source of bias:

- **Setting 1 (Initial Credit Gap):** Disparity arises from unequal initial credit score distributions ($\mathbb{E}[x_0|s^-] < \mathbb{E}[x_0|s^+]$). Repayment probabilities are taken from the fixed tables provided in the released code of Lear & Zhang (2025); we additionally re-estimate them from the Home Credit Default Risk dataset (Montoya et al., 2018) to verify consistency with the released values. See Appendix E.1 for our re-estimation procedure.

---

[1]Our adapted source code is available at: `https://anonymous.4open.science/r/causal-fair-rl`

- **Setting 2 (Adverse Repayment Model):** Similar to Setting 1, preserving the initial credit gap while deriving repayment probabilities from the Lending Club dataset (Lending Club, 2021), representing a more adverse lending environment.
- **Setting 3 (Group-dependent Drift):** Groups start with equal credit, but inequality emerges through biased drift dynamics where the disadvantaged group faces downward pressure, while using the same repayment probabilities as Setting 1.

Additional environment details are provided in Appendix D. For the full mathematical formulation of the transition dynamics, reward structure, and detailed parameters for all settings, please refer to Appendix E.

### 3.3 Training Protocol

All agents are trained with the KL-regularized PPO variants explained in section 3.1.1. We follow the original experimental configuration, and keep the architecture and hyperparameters fixed across methods: both the policy and the value functions are parameterized by multi-layer perceptrons with two fully connected hidden layers of 64 units each. For the core reproduction experiments, results are aggregated over twenty random seeds. For extension experiments, the number of seeds varies due to computational cost, and we report these results as robustness or stress-test evidence rather than direct reproductions of the original settings. To ensure comparability, we use consistent evaluation and reporting procedures across all runs and model variants.

### 3.4 Evaluation Metrics

We evaluate learned policies using the standard RL performance metric of reward-based utility and the long-term fairness metrics established by Lear & Zhang (2025).

**Utility.** In our environments the utility is defined as the agent's total expected reward, i.e. the cumulative return the lending policy achieves. In our loan setting, each approved loan contributes a reward (profit) if repaid or a negative reward (loss) if defaulted, so utility captures the overall sum of rewards and therefore the financial performance of the policy.

**Qualification Gain Disparity (QGD).** We use *qualification gain disparity* as the primary long-term fairness metric proposed by Lear & Zhang (2025). QGD measures whether one sensitive group experiences systematically larger cumulative improvements in a given metric under a learned policy. For each group, we compute the expected accumulated qualification gain over trajectories, where per-step gain is defined by a transition gain function $g(x_t, x_{t+1})$ applied to changes in, e.g., credit score. Let

$$g_{\text{total}} = \sum_{t=1}^{T} g(x_t, x_{t+1}) \tag{3}$$

denote the accumulated qualification gain over a trajectory. The disparity is defined as the difference in expected gains between the advantaged ($s^+$) and disadvantaged ($s^-$) groups:

$$C_\pi(\theta) \; = \; \mathbb{E}\left[g_{\text{total}} \mid s^+\right] \; - \; \mathbb{E}\left[g_{\text{total}} \mid s^-\right]. \tag{4}$$

$C_\pi(\theta) = 0$ indicates equal long-term qualification gains (parity).

**Causal decomposition.** To diagnose the sources of qualification gain disparity, we adopt the causal decomposition proposed by Lear & Zhang (2025). This analysis expresses the total disparity as the sum of three effects:

$$C_\pi(\theta) \; = \; \text{DPE} \; + \; \text{IPE} \; + \; \text{SPE},$$

where *DPE* (direct policy effect) captures disparity caused directly by the policy's decision rule at the time of action (immediate unequal treatment). *IPE* (indirect policy effect) captures disparity that arises through policy-induced changes in the state distribution over time (feedback dynamics). *SPE* (spurious policy effect) captures residual disparity attributable to structural or exogenous differences in the environment (e.g., unequal initial qualification or group-dependent drift).

Following Lear & Zhang (2025), we estimate these effects using two auxiliary policies: a baseline policy $\pi_0$ that always takes the negative decision, and a "virtual" policy $\pi_{\mathrm{PS}}$ that matches the state-transition dynamics of the learned policy $\pi$ while removing its direct, instantaneous effect on qualification gain. Intuitively, DPE is obtained by comparing outcomes under $\pi$ versus $\pi_{\mathrm{PS}}$ (isolating direct decision effects), IPE by comparing $\pi_{\mathrm{PS}}$ versus $\pi_0$ (isolating dynamics-driven effects), and SPE by evaluating the remaining group difference under $\pi_0$ (baseline structural disparity). In practice, we follow the authors' implementation and compute these quantities via Monte Carlo rollouts. For full derivations and the construction of $\pi_{\mathrm{PS}}$, please see Sec. 3.4 and Eqs. (3)-(4) of Lear & Zhang (2025) .

**Benefit Fairness.** Following Lear & Zhang (2025), we additionally report *benefit fairness* (Plecko & Barein-boim, 2023): individuals across groups who would benefit equally from a positive decision should receive similar treatment. We quantify violations with the pairwise loss $\Lambda(\pi)$, which penalizes cross-group differences in approval probability more strongly when the corresponding benefit gap is small; lower $\Lambda(\pi)$ indicates fairer treatment. Formal definitions of the individual benefit $\Delta(x, s)$ and of $\Lambda(\pi)$ are given in Appendix E.2.

### 3.5   Infectious Disease Environment

To verify the generalizability of PPO-C and PPO-Cb beyond the existing, restricted financial domain, we adapt the framework to an infectious disease control task. We utilize the environment implementation from Hu et al. (2023b), which models a viral spread within a social network $G = (V, E)$ using a Susceptible-Infected-Recovered (SIR) model. The disease dynamics follow a Susceptible-Infected (SI) transition where healthy individuals are infected by their neighbours with a probability $P_I$, and an infected individual recovers with probability $\rho$. The agent's objective is to allocate a limited treatment (vaccine) per time step to individuals to maximize the healthy population.

In this domain, group labels are defined structurally rather than categorically. Following Hu et al. (2023b), we utilize the *Girvan-Newman algorithm* Girvan & Newman (2002) to identify structural communities within the contact graph. These detected communities serve as the basis for evaluating long-term fairness. The fairness metric, **discrepancy** ($\delta$), measures the maximum gap in the vaccination-to-infection ratio between these groups, ensuring that resources are distributed proportionally to the epidemic burden in each faction. We also adapt **benefit fairness** ($\Lambda$) to ensure that individuals with a similar risk of infection, and thus similar potential benefit from vaccination, are treated equitably regardless of their community membership.

### 3.6   $K$-**Group Generalization**

To explore the scalability of long-term fairness constraints, we extend the original objective to scenarios with $K > 2$ sensitive groups. Details of the initial five-group lending environment are provided in Appendix C. While the original *Qualification Gain Disparity* (QGD) is defined for binary comparisons, it becomes less useful in a general multi-group context; more broadly, de-biasing with respect to a single sensitive attribute does not guarantee parity across structured subgroups defined over several protected attributes Kearns et al. (2018). Thus, we introduce **Qualification Gain Variance (QGV)** as a generalized metric. Let $\mu_k$ denote the expected cumulative qualification gain for group $s_k$,

$$\mu_k := \mathbb{E}\left[V_{do(\pi, s_k)}(x_0)\right], \quad k = 1, \dots, K \tag{5}$$

and let $\bar{\mu}$ be the mean qualification gain across groups:

$$\bar{\mu} := \frac{1}{K} \sum_{k=1}^{K} \mu_k \tag{6}$$

We define Qualification Gain Variance (QGV) as a measure of variance across groups:

$$\mathrm{QGV}(\pi) = \frac{1}{K} \sum_{k=1}^{K} \left(\mu_k - \bar{\mu}\right)^2 . \tag{7}$$

This choice is mathematically motivated directly by the original binary fairness penalty, extended to any number of groups: in the $K = 2$-group case the penalty is the squared disparity between groups, and for $K$

groups a natural analogue is the average squared disparity $\mathcal{C}_{\text{pair}} = \binom{K}{2}^{-1} \sum_{i<j} (\mu_i - \mu_j)^2$ over all unordered group pairs. As shown in Appendix G.1, this pairwise objective is proportional to the variance-based form $\mathcal{C}_{\text{pair}} = \frac{2K}{K-1}$ QGV, so minimising this objective for two groups is equivalent to minimising QGV up to a constant that can be absorbed into $\beta^C$. We therefore adopt QGV as the compact $K$-group penalty, preserving the same optimization intention as the original authors.

This variance-based construction is related in spirit to other fairness formulations that treat per-group risks as a vector to be equalized in a multi-objective sense Martinez et al. (2020) or that enforce commensurate risks across subgroups through risk and deviation measures Williamson & Menon (2019). It nonetheless targets a different object: rather than equalizing static, per-group predictive risks, QGV measures the dispersion of the expected cumulative qualification gains induced by a learned policy, thereby extending the binary QGD penalty to multiple groups while retaining the long-term, causal outcome semantics of Lear & Zhang (2025).

Correspondingly, we modify the PPO-C objective in Eq. (1) by replacing the binary squared disparity term with the variance-based $K$-group penalty:

$$J(\theta) = L^{\text{UTIL}} - \beta^{\text{KL}} L^{\text{KL}} - \beta^C \cdot \text{QGV}. \tag{8}$$

This adaptation allows the agent to simultaneously regularize multiple group-level outcomes, encouraging the policy to converge toward a state where all demographics achieve long-term parity.

### 3.7 Computational Requirements

All experiments were conducted on a high-performance computing cluster utilizing fractional nodes equipped with an NVIDIA A100 GPU (18 cores and 120 GiB of memory). Additional experiments involving infectious environments were performed locally on a MacBook Air with an M4 processor and 16 GB of RAM.

## 4 Results

In this section, we reproduce the main experimental results of Lear & Zhang (2025) and then report extensions designed to test robustness under more challenging conditions. Throughout, we present our reproduced plots in the main text, and include the corresponding plots from the original paper in Appendix B for easier comparison.

### 4.1 Results Reproducing Lear & Zhang (2025)

**Qualification Gain Disparity Comparison**: To verify Claim 1, we replicate Figure 4 of the original paper, which is designed to evaluate whether the proposed methods, PPO-C and PPO-Cb, are able to reduce QGD over time relative to standard PPO and existing fairness-aware baselines (A-PPO and F-PPO-L) across different experimental settings. As shown in Figure 1, our reproduced results are largely consistent with those reported in the original paper. In all three settings, PPO-C and PPO-Cb consistently achieve lower qualification gain disparity than PPO and other fairness-aware baselines, confirming the effectiveness of constraint-based approaches in mitigating disparity.

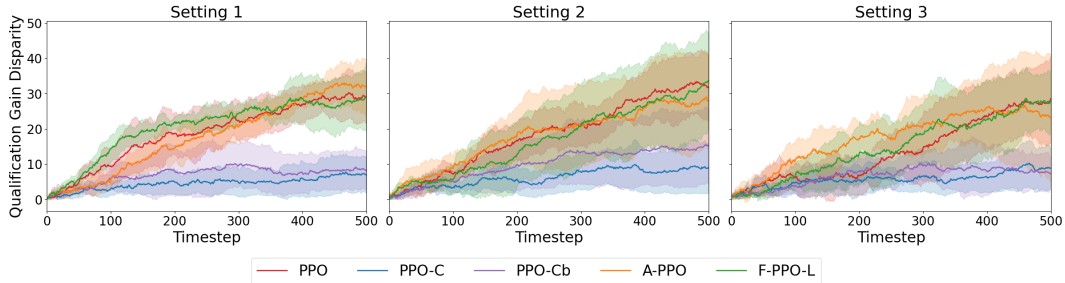

Figure 1: Qualification gain disparity comparison between the main three settings

**Utility-Fairness Trade-off**: To assess Claim 2, we reproduced the bank profit results from Figure 3 of Lear & Zhang (2025) for all five methods across the three settings. Figure 2 shows that PPO-C and PPO-Cb do not consistently achieve "competitive" long-term utility under fairness constraints, in contrast to the original paper's reported trade-off. PPO-C performs well in Setting 3 but is substantially worse than all other methods in Setting 2. PPO-Cb is closer to PPO-C in Setting 2 and performs reasonably in Settings 2-3, but underperforms in Setting 1 relative to fairness-aware baselines such as F-PPO-L. Overall, these utility results are weaker than those reported in the original study, and we do not reproduce the competitive performance of PPO-C/PPO-Cb in Setting 2.

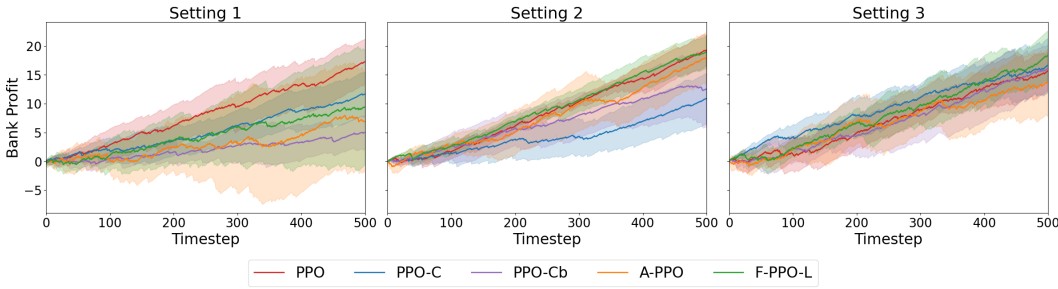

Figure 2: Utility comparison between the main three settings

**Benefit Fairness and Source of Disparity Reduction**: To evaluate the part of claim 3 stating that PPO-Cb primarily reduces DPE relative to PPO-C, we reproduce the causal decomposition of QGD. Figure 3 shows how these three components, discussed in section 3.4, behave during training, together with plotting the total disparity $C_\pi(\theta)$. This graph corresponds with Figure 5 of Lear & Zhang (2025). DPE exhibits larger fluctuations than IPE across both settings, particularly during early training, where DPE drops sharply while IPE remains relatively stable. In addition, when using PPO-Cb, DPE stabilizes more noticeably in the mid-to-late training phase. These results align with the original paper's interpretation that the benefit-fairness regularizer $\beta_\Lambda \Lambda$ reduces disparity mainly through DPE. This result also shows that short-term/instantaneous effects are still crucial for long-term fairness.

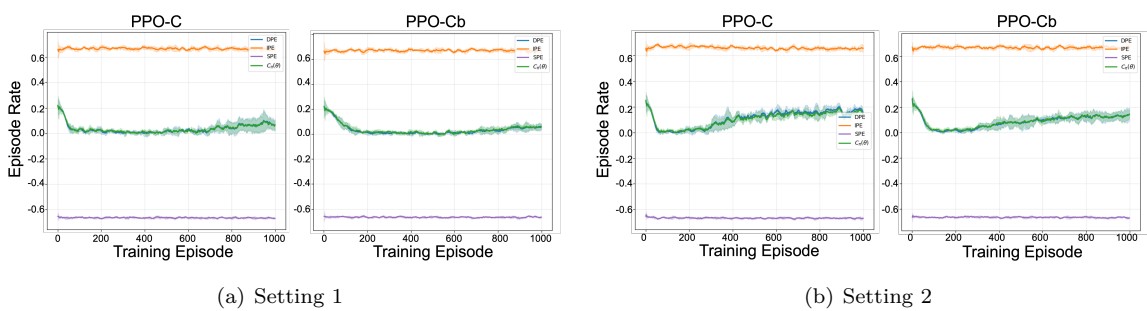

(a) Setting 1                                  (b) Setting 2

Figure 3: Qualification Gain Disparity decomposition for PPO-C and PPO-Cb settings 1 and 2

To evaluate the second part of Claim 3, whether PPO-Cb improves benefit fairness, we reproduce the benefit-fairness experiment (Fig. 6 of Lear & Zhang, 2025). Since the "small" and "large" $\beta^\Lambda$ values were not specified in the paper or code, we use the values provided by the original authors upon request: $\beta^\Lambda = 1.0$ ("small") and $\beta^\Lambda = 3.0$ ("large") for Setting 1, and $\beta^\Lambda = 1.0$ ("small") and $\beta^\Lambda = 5.0$ ("large") for Setting 2, with $\beta^C = 0.65$ and $\beta^C = 0.75$ respectively. Our reproduced curves, seen in Figure 4, closely match the original: PPO-Cb achieves lower benefit-fairness loss $\Lambda(\pi)$ than PPO-C , indicating more similar treatment of cross-group individuals with comparable benefit. Together with the decomposition results above, this supports the link between benefit-fairness regularization and reductions in DPE: enforcing similar treatment for similarly benefiting individuals primarily constrains the policy's *instantaneous* decision bias, which then translates into lower long-run qualification gain disparity.

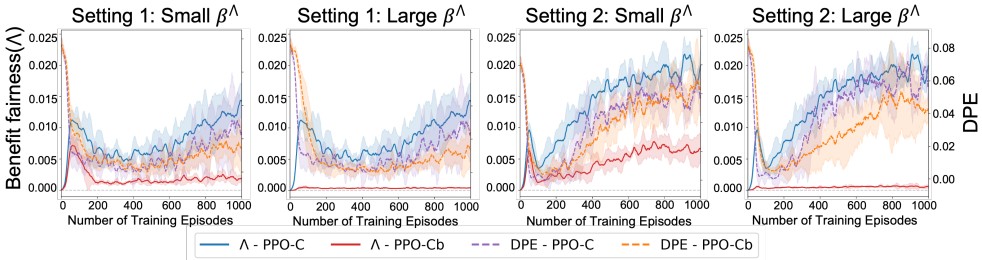

Figure 4: Benefit fairness and DPE for PPO-C and PPO-Cb model variants

## 4.2 Results Beyond Lear & Zhang (2025)

**Comparison of QGD Across Five Methods:** To provide a more comprehensive comparison of PPO-C and PPO-Cb with other baselines, we perform the decomposition analysis across the other trained models. The corresponding results are shown in Figure 5.

The comparison shows that F-PPO-L, PPO, and A-PPO, after experiencing initial fluctuations, all converge to relatively high positive values of qualification gain disparity, indicating that unfairness is not effectively mitigated by these methods. When considered together with the results in Figure 3, it can be observed that PPO-C and PPO-Cb not only maintain acceptable long-term utility, but also begin to effectively control unfairness at early stages of training. This suggests that, compared to other methods, PPO-C and PPO-Cb introduce explicit causal fairness constraints that constrain policy gradients from the outset, leading to early suppression of disparity and improved effectiveness and stability in fairness control.

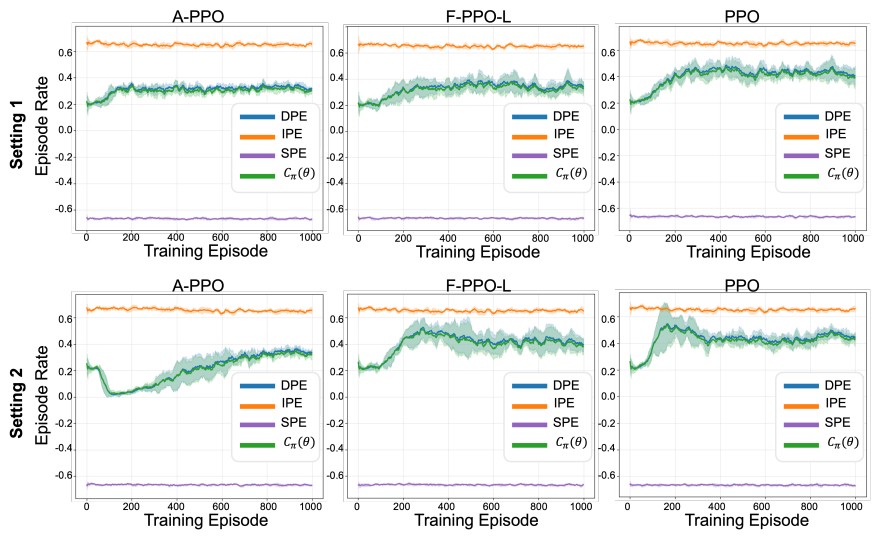

Figure 5: QGD decomposition of other 3 methods in settings 1 and 2

**Long-term Fairness Dynamics in an Infectious Disease Environment:** To evaluate generalization beyond financial domains, we tested PPO-C in the infectious disease environment against standard baselines and two heuristics: MAX (greedy) and RANDOM. As shown in Figure 6, PPO-C successfully mitigates long-term disparity but faces a severe utility trade-off.

In terms of fairness, PPO-C outperforms unconstrained baselines. While A-PPO achieves the best short-term fairness (Figure 6(a)), PPO-C excels in long-term fairness (Figure 6(b)), matching F-PPO with a discrepancy score of $\approx 3.0$, significantly lower than PPO, A-PPO, and MAX (ranging 4.5-7.0). RANDOM achieves the lowest long-term disparity, as it inherently avoids systematic bias.

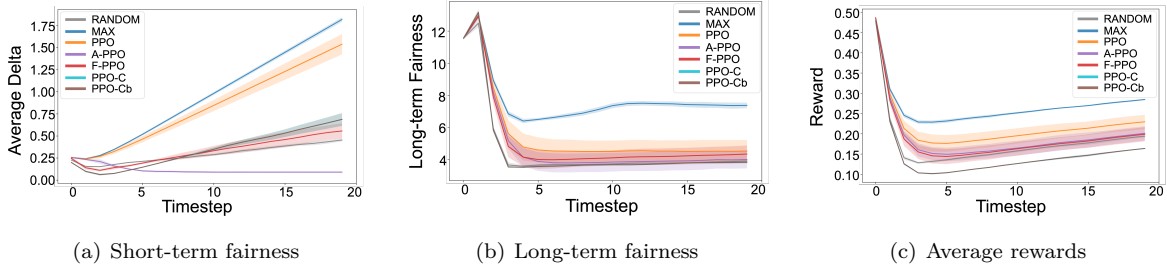

(a) Short-term fairness     (b) Long-term fairness     (c) Average rewards

Figure 6: Experimental results for epidemic control

However, this equity incurs a high cost. As seen in Figure 6(c), PPO-C yields the lowest average reward (0.16), underperforming even the RANDOM policy (0.19), whereas the greedy MAX policy achieves the highest utility (0.28). This indicates that in environments where optimal utility requires concentrated interventions (e.g., targeting super-spreaders), PPO-C's strict fairness constraints may overwhelm the reward signal, hindering effective epidemic control.

**Balancing Utility and Long-term Fairness under Group Imbalance:** We investigate robustness to demographic imbalance by varying Group 0's population ratio to 0.3 and 0.7 across all settings. PPO-C and PPO-Cb demonstrate superior generalization, maintaining disparity close to zero in all configurations while baselines like A-PPO and F-PPO-L exhibit high volatility. Crucially, this stability does *not* come at the cost of degenerate utility. Both methods maintain non-negative profits, and in Setting 3 (ratio 0.3), PPO-C approaches the performance of unconstrained PPO. These results confirm the methods generalize reliably to demographically skewed environments. All results for these settings can be found in Appendix A.

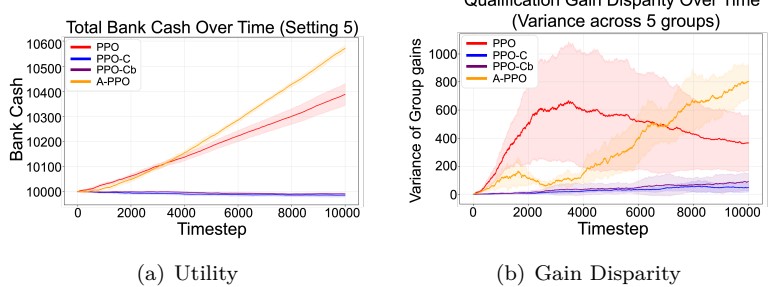

(a) Utility        (b) Gain Disparity

Figure 7: Initial untuned $K = 5$ QGV experiment. QGV reduces gain variance, but the tested coefficient causes severe utility loss, motivating the beta-sensitivity analysis.

**Generalization to $K$-Group Settings:** To evaluate whether the causal fairness objective scales beyond binary sensitive groups, we extend the lending environment to $K > 2$ groups and replace binary Qualification Gain Disparity (QGD) with Qualification Gain Variance (QGV). The endpoint-only experiment in Appendix C.3 motivates this change: constraining only the most advantaged and most disadvantaged groups can produce deceptively low endpoint disparity while leaving intermediate groups unconstrained.

Our initial $K = 5$ QGV experiment shows a clear failure mode. As shown in Figure 7, QGV can drive qualification-gain variance close to zero, but the tested coefficient also causes bank cash to remain nearly flat. This indicates that the policy reduces disparity by suppressing lending rather than by improving qualification gains across all groups.

However, follow-up ablation experiments show that this is not an inherent failure of QGV. Figure 8 compares QGV policies across $\beta$ values and group counts $K$. High or default $\beta$ values often reduce QGV at large utility

cost, whereas moderate beta values, especially around $\beta^C = 0.05$, frequently preserve most of PPO's utility while reducing qualification-gain variance.

Figure 9 summarizes the best moderate-QGV policy for each setting and group count. The results are setting-dependent rather than monotonically worse as $K$ increases: Setting 2 with $K = 4$ is the strongest case, while Setting 1 with $K = 3$ and Setting 2 with preliminary $K = 5$ remain weak or uncertain. Finally, Figure 10 shows that QGV should be evaluated through per-group gains, not only aggregate variance, since a low variance score can still be produced by uniformly suppressing gains.

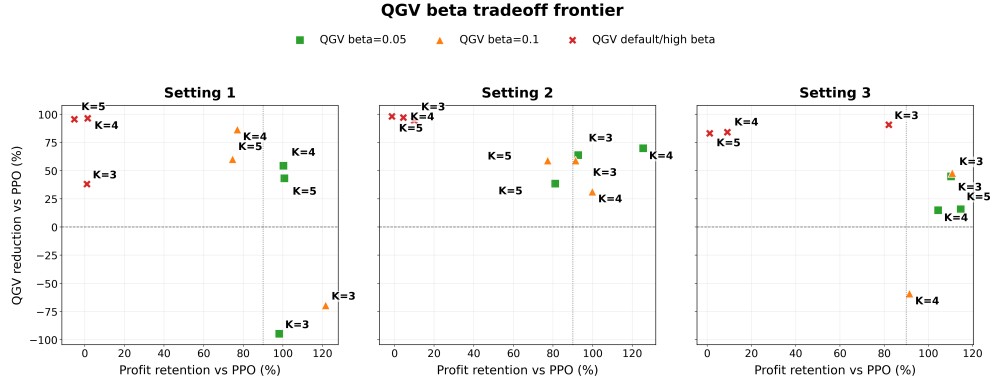

Figure 8: Utility–fairness trade-off of QGV across beta values for $K = 3, 4, 5$. Points in the upper-right preserve more PPO utility while reducing more QGV.

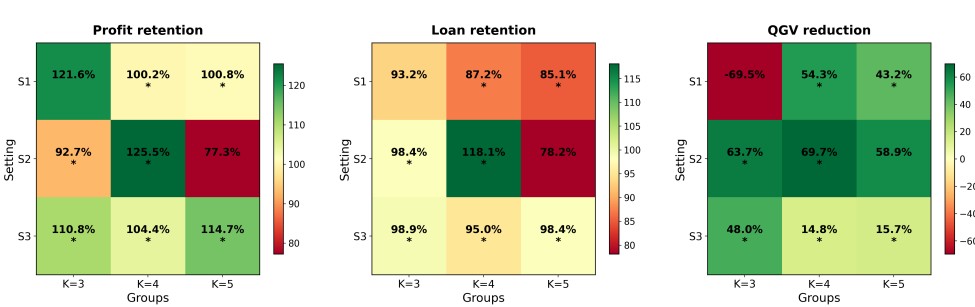

Figure 9: Best moderate-QGV result for each setting and group count. Moderate QGV often preserves utility while reducing qualification-gain variance, but the result varies across settings.

**Causal Decomposition of $K$-Group QGV:** To test whether the causal diagnostic remains useful beyond binary groups, we recomputed the group-level causal components for saved multi-group checkpoints and aggregated them through QGV. For each group $k \in \{0, 1, ..., K - 1\}$, we estimate the structural, indirect, and direct gain components $(S_k, I_k, D_k)$ using the same baseline and path-specific rollouts as in the binary QGD decomposition, then compute the ordered variance decomposition described in Appendix G.2. We report the pair-normalized quantity $\text{QGV}_{pair} = \frac{2K}{K-1}\text{QGV}$ so that values are comparable across different $K$ and reduce to squared QGD when $K = 2$.

Figure **??** shows that QGV is strongly coefficient-sensitive. In the $K = 5$, Setting 1 sweep, very small penalties behave similarly to PPO, while moderate $\beta^C$ values reduce QGV while preserving positive utility. Larger penalties continue reducing QGV but induce low-lending behavior and utility collapse.

The decomposition clarifies the mechanism: across policies, the SPE is positive and large, while the IPE is negative with nearly equal magnitude, meaning that the state-distribution component offsets much of the

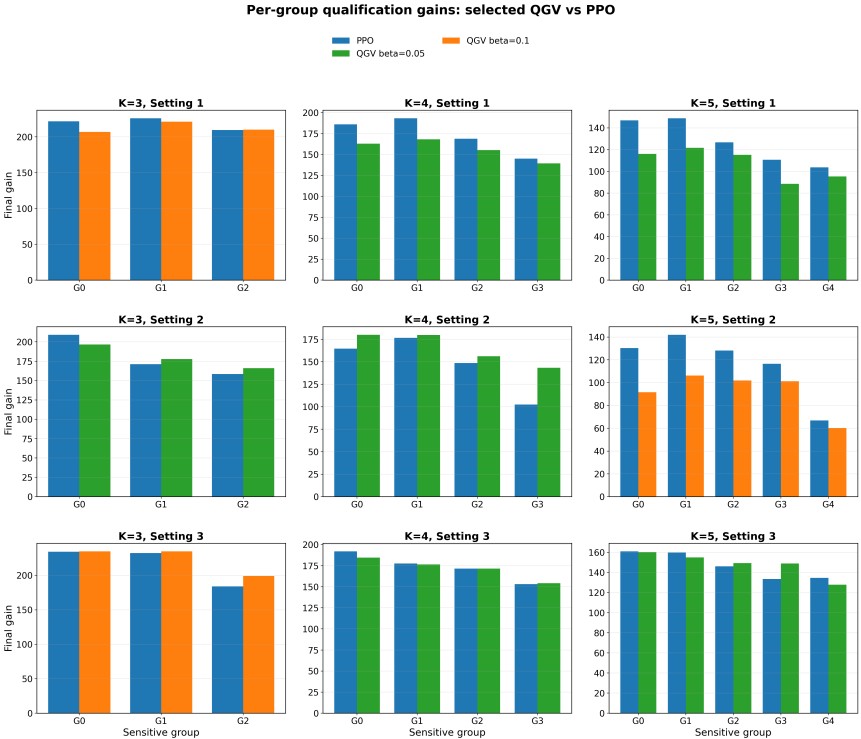

Figure 10: Per-group qualification gains for PPO and the selected moderate-QGV policy. This diagnostic checks whether QGV equalizes gains across all groups rather than only reducing the aggregate variance metric.

structural variance. The remaining total QGV is therefore driven mainly by the DPE. As $\beta^C$ increases, QGV-DPE shrinks toward zero. Thus, in our experiments, PPO-C with QGV mainly suppresses direct policy-induced residual disparity rather than eliminating structural disparity.

## 5 Discussion

### 5.1 Assessment of Claims

Our study confirms the core algorithmic contributions of Lear & Zhang (2025) while identifying specific limitations for their application.

**Claims 1 & 3: Effectiveness and Mechanism.** The consistent reduction of QGD across all replication and extension settings (balanced and unbalanced populations) validates the causal framework's efficacy. Crucially, our decomposition results support the authors' theoretical link between *benefit fairness* and the *Direct Policy Effect*. By regularizing treatment parity for individuals with similar potential gains, PPO-Cb successfully stabilizes the policy's immediate impact, which serves as the primary lever for long-term parity.

**Claim 2: The Utility-Fairness Frontier.**

While we generally support the claim of competitive utility, our reproduction of Setting 2 revealed a more sensitive trade-off than originally reported, and the $K$-group QGV extension refines this further. With an untuned or overly large coefficient, the QGV penalty can dominate the reward signal and induce a degenerate low-lending policy. This behavior is coefficient-dependent rather than intrinsic to QGV: moderate $\beta^C$ values recover meaningful utility–fairness trade-offs in several $K > 2$ settings, suggesting the framework remains viable beyond binary groups but requires careful calibration as $K$ grows.

**Claim 4: Sources of Disparity.**

Our reproduced binary decompositions confirm that reductions in QGD are primarily driven by DPE. The new QGV decomposition extends this diagnostic to $K$-group settings. After decomposing each group-level gain and aggregating through variance, we again find that the residual multi-group disparity is mainly controlled through the direct policy component: SPE remains large, IPE often offsets it, and increasing $\beta^C$ mostly shrinks DPE. This supports Claim 4 beyond the binary case, while also showing a limitation: causal policy regularization can prevent the learned policy from adding disparity, but it does not remove structural disparity in the environment.

## 5.2 Limitations and Generalizability

The extension experiments suggest that the robustness of PPO-C and PPO-Cb depends on the environment. In the lending experiments with population imbalance, the methods maintain low disparity without collapsing utility, indicating that original fairness objective can tolerate moderate changes to group proportions. However, this behaviour does transfer to the infectious-disease environment. In that task, vaccinating one node also affects the infection risk of neighbouring nodes, so high utility may require concentrating treatment on structurally important individuals. This can conflict with group-level fairness constraints. In our experiments PPO-C improves long-term fairness but obtains the lowest reward of any method we tested, including the RANDOM baseline. This indicates that the utility cost of causal fairness constraints depends on whether fair actions and high-utility actions are aligned in the environment.

Furthermore, the multi-group extension shows that QGV is more sensitive to coefficient scaling than binary QGD. Default or high $\beta^C$ values can suppress lending and collapse utility, while moderate values can recover more useful utility–fairness trade-offs. Future work should therefore investigate adaptive scaling rules for $\beta^C$ as a function of the number of groups, the magnitude of QGV, and the variance of group-level qualification gains.

## 5.3 Environmental Impact

In total, we used approximately 368 A100 GPU-hours plus $\sim$2 CPU-hours, leading to estimated emissions of $\approx$51 kgCO$_2$eq, directly offset by the compute provider. Details can be found in Appendix F.

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

# A    Utility and QGD comparison under Unbalanced Group Populations

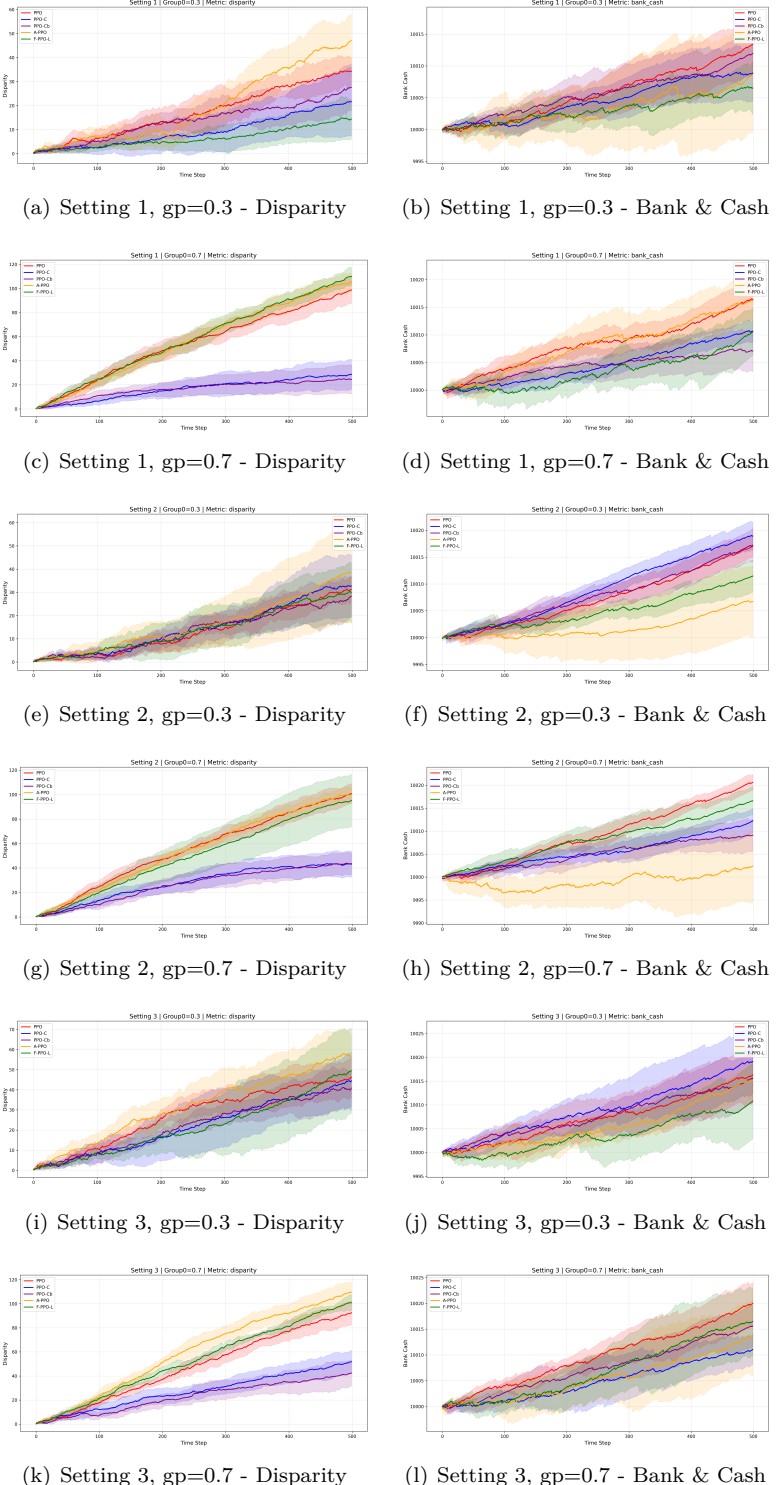

(a) Setting 1, gp=0.3 - Disparity      (b) Setting 1, gp=0.3 - Bank & Cash

(c) Setting 1, gp=0.7 - Disparity      (d) Setting 1, gp=0.7 - Bank & Cash

(e) Setting 2, gp=0.3 - Disparity      (f) Setting 2, gp=0.3 - Bank & Cash

(g) Setting 2, gp=0.7 - Disparity      (h) Setting 2, gp=0.7 - Bank & Cash

(i) Setting 3, gp=0.3 - Disparity      (j) Setting 3, gp=0.3 - Bank & Cash

(k) Setting 3, gp=0.7 - Disparity      (l) Setting 3, gp=0.7 - Bank & Cash

Figure 11: Utility and qualification gain disparity comparison under unbalanced groups

# B Comparison with original paper

For each reproduced result in Section 4, and for the sake of clear comparison, we include the corresponding plots from Lear & Zhang (2025).

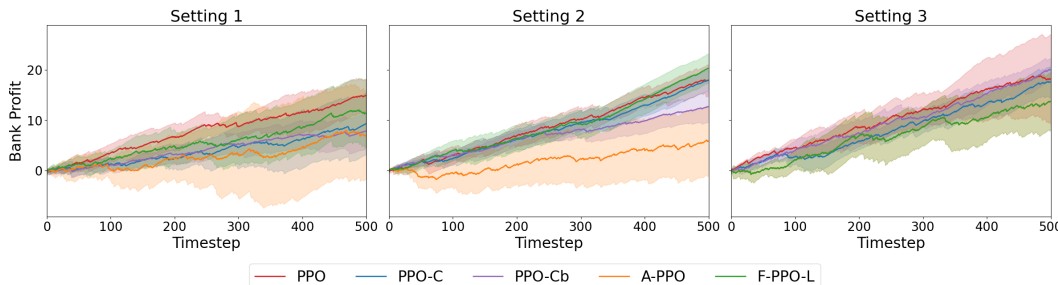

Figure 12: Original paper's utility (bank profit) comparison across experimental settings (Lear & Zhang, 2025, Fig. 3)).

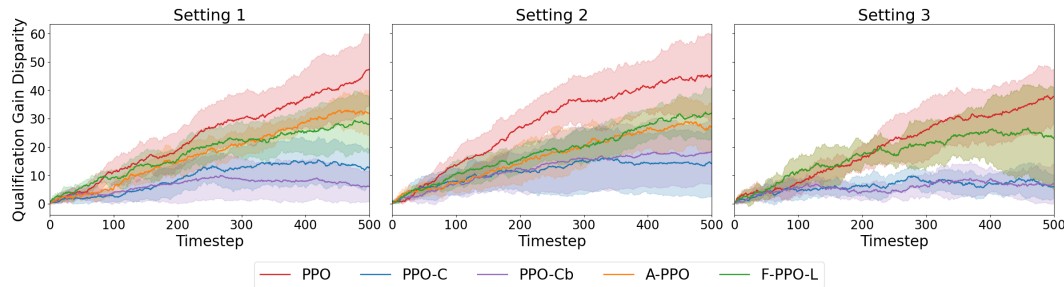

Figure 13: Original paper's qualification gain disparity comparison across experimental settings (Lear & Zhang, 2025, Fig. 4)).

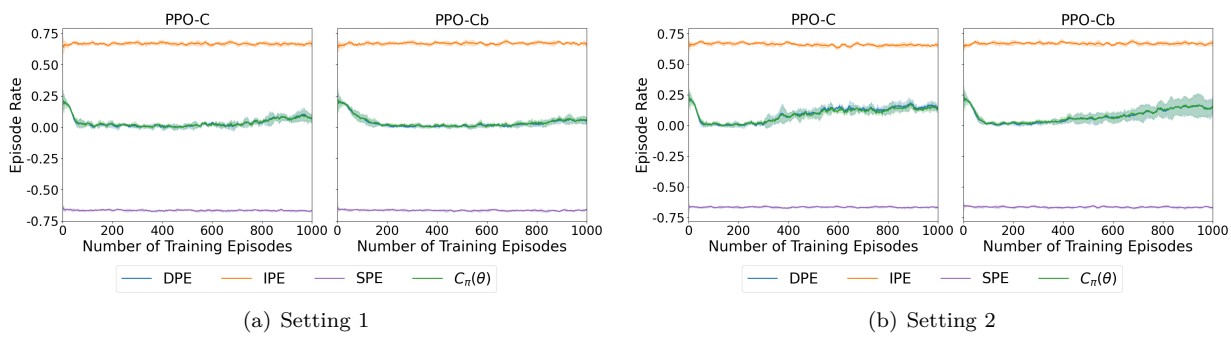

Figure 14: Original paper's causal decomposition of qualification gain disparity for PPO-C and PPO-Cb in Settings 1 and 2 (Lear & Zhang, 2025, Fig. 5)).

# C Generalization to $K$-Group Environments

To evaluate the scalability and generalization of the methods proposed by Lear & Zhang (2025), we extend the binary group setting to an environment with five sensitive groups ($K = 5$).

### C.1 Settings for 5 Groups

In this setting (Setting 5), we move beyond binary disparities to a gradual shift in credit distributions. This simulates a more complex social structure where multiple groups occupy an intermediate spectrum between the most disadvantaged and most advantaged populations.

**Group Likelihoods:** The population is partitioned equally across the five groups, with each group having a uniform likelihood of being sampled:

$$P(s_k) = 0.2, \quad \text{for } k \in \{0, 1, 2, 3, 4\}. \tag{9}$$

**Initial Credit Distributions:** The initial credit score distributions are manually synthesized to represent a progression. Groups 0 and 1 represent the original "disadvantaged" profile, Group 2 represents an intermediate state, and Groups 3 and 4 represent the original "advantaged" profile. The specific probability vectors for the 14 credit clusters are detailed in Table 1.

Table 1: Initial Credit Cluster Probabilities (Setting 5)

| Group | C1 | C2 | C3 | C4 | C5 | C6 | C7 | C8 | C9 | C10 | C11 | C12 | C13 | C14 |
|---|---|---|---|---|---|---|---|---|---|---|---|---|---|---|
| Group 0 | 0.0 | 0.0 | .05 | .05 | .05 | .05 | 0.1 | 0.1 | .15 | .15 | .15 | .15 | 0.0 | 0.0 |
| Group 1 | 0.0 | 0.0 | .05 | .05 | .05 | .05 | 0.1 | 0.1 | .15 | .15 | .15 | .15 | 0.0 | 0.0 |
| Group 2 | .025 | .025 | .05 | .05 | .075 | .075 | .125 | .125 | .15 | .15 | .075 | .075 | 0.0 | 0.0 |
| Group 3 | .05 | .05 | .05 | .05 | 0.1 | 0.1 | .15 | .15 | .15 | .15 | 0.0 | 0.0 | 0.0 | 0.0 |
| Group 4 | .05 | .05 | .05 | .05 | 0.1 | 0.1 | .15 | .15 | .15 | .15 | 0.0 | 0.0 | 0.0 | 0.0 |

**Drift Dynamics:** To capture different rates of environmental credit evolution, we also assign group-specific drift probabilities. These probabilities $(P_{\text{down}}, P_{\text{stay}}, P_{\text{up}})$ represent the likelihood of a credit score decreasing, staying the same, or increasing due to exogenous factors. The group-specific drift probabilities are shown in Table 2.

Table 2: Group-Specific Drift Probabilities (Setting 5)

| Group | Down $(-1)$ | Stay $(0)$ | Up $(+1)$ |
|---|---|---|---|
| Groups 0 & 1 | 0.10 | 0.80 | 0.10 |
| Group 2 | 0.08 | 0.84 | 0.08 |
| Groups 3 & 4 | 0.05 | 0.90 | 0.05 |

### C.2 Performance Analysis

We evaluate the 5-group setting using the *Qualification Gain Variance* (QGV) objective. Figure 15 shows the cumulative qualification gain for each group separately to further depict how different algorithms tackle this setting.

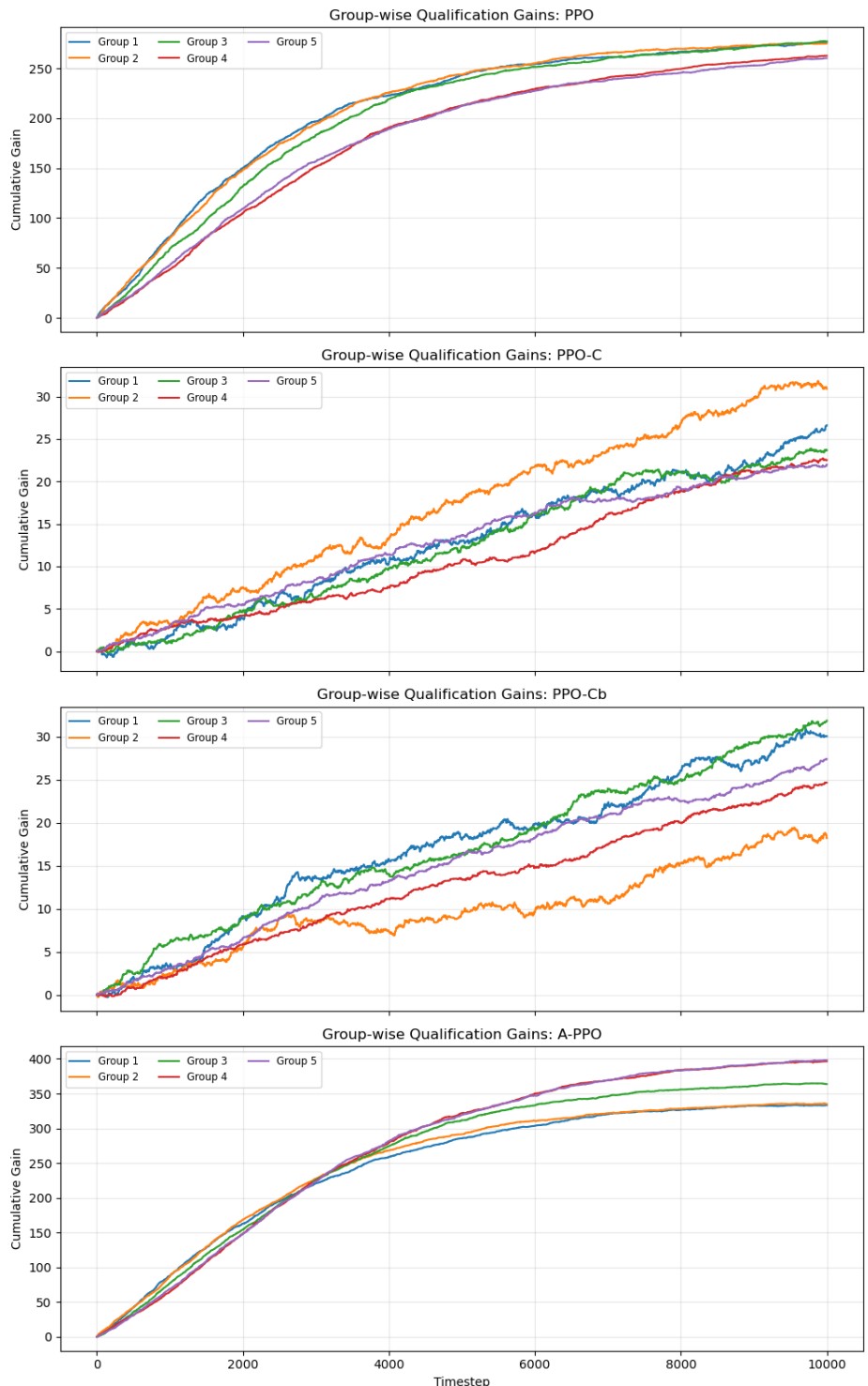

Figure 15: Cumulative qualification gain per group for each model variant.

## C.3 Endpoints analysis

As an initial $K$-group extension, we trained the PPO-C model, along with a fairness-aware baseline (A-PPO) and plain PPO, in the $K = 5$ environment by applying them only to the *endpoint* groups. Concretely, we

set `GROUP_MIN_ID = 0` and `GROUP_MAX_ID = 4` and computed the fairness terms using only trajectories from these two groups; intermediate groups (1-3) were not directly constrained by the fairness regularizers.

Under this protocol, the reported qualification gain disparity corresponds to an extreme-group gap,

$$\text{QGD}^{(4-0)} \;=\; \mathbb{E}[g_{\text{total}} \mid s = 4] \;-\; \mathbb{E}[g_{\text{total}} \mid s = 0]\,,$$

rather than a multi-group notion of parity. Likewise, the benefit-fairness loss $\Lambda(\pi)$ and the causal decomposition terms are computed only for the pair $(s = 0, s = 4)$. This experiment should therefore be interpreted as testing whether the original methods can reduce disparity between the most-advantaged and most-disadvantaged groups and whether this then translates into a beneficial policy for all groups or only the endpoints.

Figure 16 summarizes aggregate utility (bank profit) and endpoint QGD. Under these aggregate metrics, PPO-C appears strong in both settings, attaining low disparity while managing better utility than A-PPO. However, the per-group qualification-gain trajectories in Figures 17-18 show that this improvement is largely driven by suppressing the cumulative gains of the endpoint groups (0 and 4), rather than fairly) improving outcomes across all groups. A-PPO illustrates an even more extreme scenario in Setting 2: it achieves low QGD, by simply denying nearly all loans to groups 0 and 4, while still obtaining good utility from intermediate groups. In Setting 1, A-PPO behaves similarly to PPO in disparity but yields the weakest utility among the compared methods.

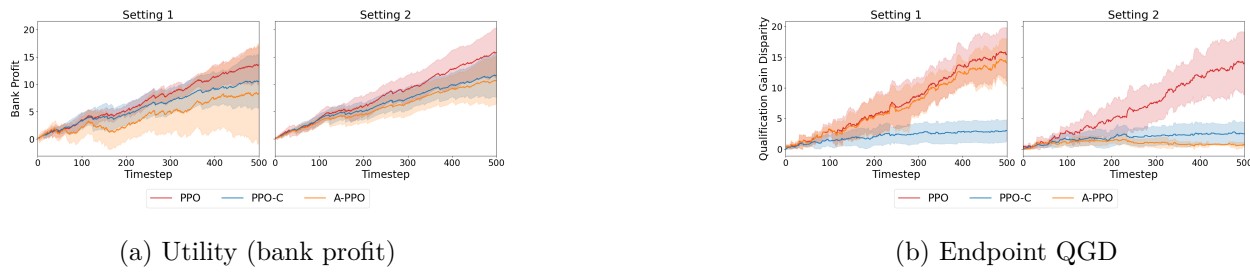

(a) Utility (bank profit)                (b) Endpoint QGD

Figure 16: Aggregate metrics for the $K = 5$ endpoints experiment ($s = 4$ minus $s = 0$).

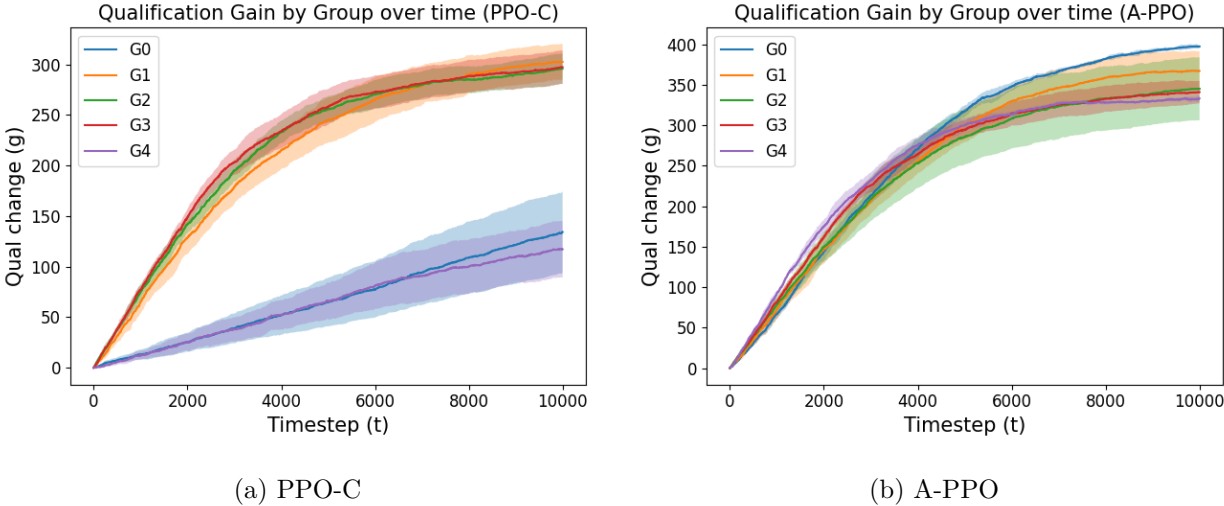

(a) PPO-C                (b) A-PPO

Figure 17: Cumulative qualification gain per group over training in Setting 1 (endpoints protocol).

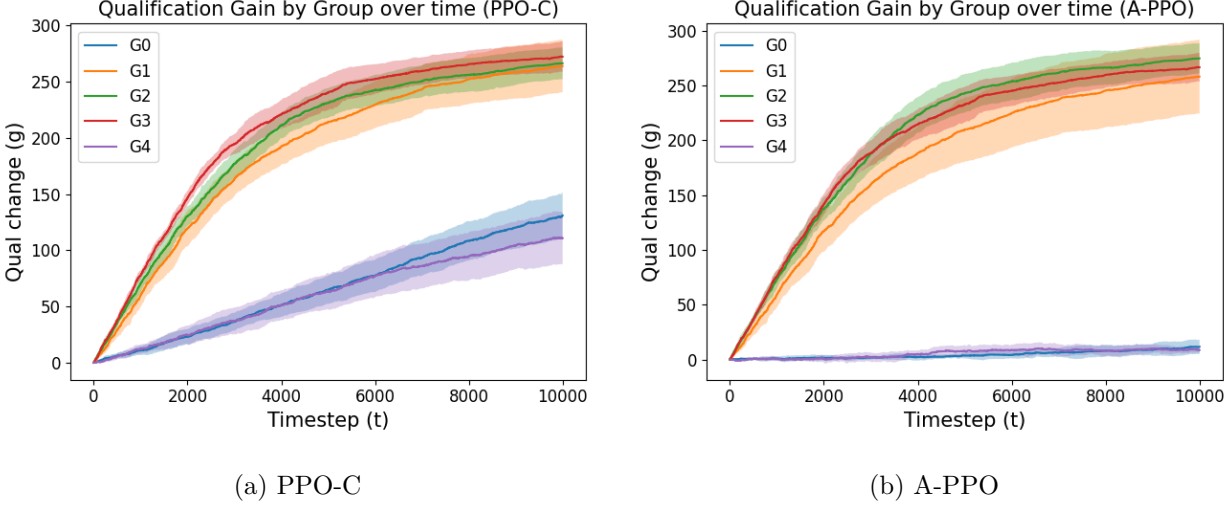

(a) PPO-C  (b) A-PPO

Figure 18: Cumulative qualification gain per group over training in Setting 2 (endpoints protocol).

## D    Infectious Disease Environment Details

The infectious disease experiments are conducted using a graph-based SIR model where the population structure is determined by the **Zachary's Karate Club graph** (34 nodes, 78 edges). The environment is initialized with a single random infection. The parameters for the environment are set following the baseline paper (Hu et al., 2023b):

- **Transition Probabilities**: Infection probability $\tau = 0.5$ per contact; recovery probability $\rho = 0.005$ per timestep.

- **Treatment Capacity**: The agent is limited to 1 vaccination per timestep ($V_{\max} = 1$), which induces an immediate transition to the recovered state.

- **Community Partitioning**: We apply the Girvan-Newman algorithm to the graph, using the first detected split to define two primary communities ($s^+$ and $s^-$) as the sensitive groups.

- **Reward Function**: The utility for the agent is the percentage of healthy individuals in the population at each step.

## E    Lending Environment Details

The sequential lending environment is implemented as a semi-synthetic MDP based on parameters derived from the original study (Lear & Zhang, 2025). The core parameters used across all three settings are summarized below:

- **State Space**: The credit score $x_t \in \{1, 2, \ldots, 14\}$, representing discretized credit clusters.

- **Reward Function**:
    - Repayment ($y = 1$): $+1$ reward (profit).
    - Default ($y = 0$): $-10$ reward (loss from non-repayment).
    - Loan Denial ($d = 0$): $0$ reward.

- **Transitions**:
    - If denied ($d = 0$): $x_{t+1} = \text{clip}(x_t + x^{\text{drift}}, 1, 14)$.

      – If approved ($d = 1$): $x_{t+1} = \text{clip}(x_t + x^{\text{drift}} + [2y - 1], 1, 14)$.

- **Repayment Probabilities**: Fixed probability vectors $P(y = 1|x)$ derived from the Home Credit Default Risk dataset (used in Settings 1 & 3) and the Lending Club dataset (used in Setting 2).

- **Drift Probabilities**: For Settings 1 and 2, $P(x^{\text{drift}})$ is group-invariant with $P(-1) = 0.1, P(0) = 0.8, P(1) = 0.1$. In Setting 3, Group $s^-$ has $P(-1) = 0.15$ and $P(1) = 0.05$ to simulate downward pressure.

- **Initial Distribution**:
  - *Population Ratio*: Split 50/50 between $s^+$ and $s^-$ (except in the group imbalance extension).
  - *Credit Scores*: In Settings 1 and 2, $s^-$ is initialized with a lower expected credit score distribution than $s^+$. In Setting 3, both groups share the same initial distribution.

### E.1   Re-estimating repayment probabilities from data

For completeness, we produced a script that re-estimates repayment (non-default) probabilities from the Home Credit and Lending Club datasets via quantile binning, matching the simulator's discretization into 14 credit clusters. For Home Credit, we use `EXT_SOURCE_2` as the score and `TARGET` as the default label; for Lending Club, we map `sub_grade` to an ordinal score and derive a binary default label from `loan_status`. We then compute the default rate per quantile bin and report the corresponding repayment probabilities ($1 -$ default_rate). The resulting probabilities were very close to those reported by Lear & Zhang (2025); to keep results maximally comparable to the original experiments, we therefore use the paper's fixed probabilities (found in their README.md) in all main runs.

### E.2   Benefit Fairness

As stated in part 3.4, we report *benefit fairness* as an outcome-based notion of fairness, proposed by Plecko & Bareinboim (2023). This section serves as to further explain what this metric entails. Benefit fairness requires that individuals from different sensitive groups receive similar probabilities of the positive decision whenever they would benefit equally from that decision. Following Lear & Zhang (2025), the (individual) *benefit* of treatment is defined as the expected increase in qualification gain from approving ($d_1$) rather than denying ($d_0$):

$$\Delta(x, s) \;=\; \sum_{x'} \Big( P(x' \mid x, d_1, s) - P(x' \mid x, d_0, s) \Big) g_s(x, x'),$$

where $d_1$ denotes the positive decision, $d_0$ the negative decision, and $g_s(x, x')$ is the qualification gain accrued when transitioning from $x$ to $x'$ for group $s$. Benefit fairness can be stated as: for any $(x, s^+)$ and $(x', s^-)$, if $\Delta(x, s^+) = \Delta(x', s^-)$, then $\pi(d_1 \mid x, s^+) \approx \pi(d_1 \mid x', s^-)$.

To quantify violations, we use the pairwise *benefit fairness loss* from Lear & Zhang (2025):

$$\Lambda(\pi) \;=\; \sum_{x,x'} \epsilon \cdot \frac{\big|\pi(d_1 \mid x, s^+) - \pi(d_1 \mid x', s^-)\big|}{\epsilon + \big|\Delta(x, s^+) - \Delta(x', s^-)\big|} \, P(x \mid s^+) P(x' \mid s^-).$$

Here, $\epsilon > 0$ is a small constant that prevents division by zero and sets the tolerance within which we enforce similar *approval probabilities* (i.e., $\pi(d_1 \mid x, s^+) \approx \pi(d_1 \mid x', s^-)$ when $|\Delta(x, s^+) - \Delta(x', s^-)| \lesssim \epsilon$). Lower values of $\Lambda(\pi)$ indicate better benefit fairness. Since PPO-Cb explicitly regularizes this objective, it is expected to achieve the best performance of the models.

## F   Computational requirements and environmental impact details

All experiments, excluding the evaluations on alternative environments, were conducted on a high-performance computing cluster node on a national supercomputer. The specific node configuration used was a GPU-enhanced compute node (gcn) equipped with **four NVIDIA A100 Tensor Core GPUs** (40GB

HBM2 VRAM each, TDP of 400W) and two **Intel Xeon Platinum 8360Y** processors. The experiments on alternative environments were conducted on a **MacBook Air with an M4 chip**.

In total, our replication and extension experiments required approximately **368 hours of GPU computation** and **2 hours of CPU computation**. The breakdown is as follows:

- **Core Replication:** Settings 1, 2, and 3 across twenty seeds required **128 hours** of GPU computation.

- **Group Imbalance Extension:** Evaluating demographic disparity (Group0 ratios 0.3 and 0.7) required **64 hours** of GPU computation.

- **Ablation Study:** Investigating $\beta^{\Lambda}$ sensitivity accounted for **48 hours** of GPU computation.

- $K$**-Group Extension:** Testing the Qualification Gain Variance (QGV) metric including basic experiments and hyperparameter search took **128 hours** of GPU computation.

- **Alternative Environments:** Preliminary tests on the M4 chip took **2 hours**.

### F.1 CO2 Emission Related to Experiments

Experiments were conducted using NVIDIA A100 GPUs (400W TDP). A cumulative total of 368 GPU hours was consumed. Carbon emissions were estimated using a carbon efficiency of 0.35 kgCO$_2$eq/kWh.

The total emissions are estimated to be **50.87 kgCO$_2$eq**. To put this into perspective, this is equivalent to:

- Driving an average passenger vehicle for approximately **206 kilometers** (129 miles);

- The carbon footprint of producing and consuming approximately **23 beef hamburgers**;

- The energy required to manufacture roughly **616 standard 500-milliliter plastic (PET) bottles**.

Of these emissions, **100 percent** were directly offset by the cloud provider. Estimations were conducted using the Machine Learning Impact calculator presented in Lacoste et al. (2019).

## G  Generalization from Qualification Gain Disparity to Qualification Gain Variance

### G.1  Generalizing Qualification Gain Disparity to Multiple Groups

In the binary setting, long-term fairness is captured by the *qualification gain disparity* (QGD), defined as the difference in expected cumulative qualification gain between two groups:

$$\mathrm{QGD}_{1,2}(\pi) = \mathbb{E}[V_{do(\pi, s_1)}(x_0)] - \mathbb{E}[V_{do(\pi, s_2)}(x_0)].$$

Let us define, more generally, the group-level expected qualification gain:

$$G_k(\pi) := \mathbb{E}[V_{do(\pi, s_k)}(x_0)], \quad k = 1, \dots, K.$$

Then in the binary case, $\mathrm{QGD}_{1,2}(\pi) = G_1(\pi) - G_2(\pi)$, and complete fairness corresponds to $G_1(\pi) = G_2(\pi)$.

### G.2  Causal decomposition of QGV

The binary QGD decomposition cannot be applied directly to the scalar QGV after squaring and averaging, because variance introduces interaction terms between components. We therefore first decompose each group's expected qualification gain and then aggregate through QGV.

Let $G_k^0$, $G_k^{PS}$, and $G_k^\pi$ denote the expected qualification gain for group $k$ under the baseline policy, the path-specific virtual policy, and the learned policy respectively. Define

$$S_k = G_k^0, \tag{10}$$

$$I_k = G_k^{PS} - G_k^0, \tag{11}$$

$$D_k = G_k^\pi - G_k^{PS}. \tag{12}$$

Then $G_k^\pi = S_k + I_k + D_k$. For any vector $v \in \mathbb{R}^K$, define

$$Q(v) = \frac{1}{K} \sum_{k=1}^K (v_k - \bar{v})^2.$$

The ordered QGV decomposition is

$$\text{QGV-SPE} = Q(S), \tag{13}$$

$$\text{QGV-IPE} = Q(S + I) - Q(S), \tag{14}$$

$$\text{QGV-DPE} = Q(S + I + D) - Q(S + I). \tag{15}$$

Hence

$$\text{QGV}_{total} = Q(S + I + D) = \text{QGV-SPE} + \text{QGV-IPE} + \text{QGV-DPE}.$$

Unlike raw variance terms, the ordered IPE and DPE terms may be negative; a negative value means that the corresponding causal component reduces variance across groups.

### G.3 $K$-Group extension

For $K > 2$, the natural generalization of qualification gain parity requires

$$G_1(\pi) = G_2(\pi) = \cdots = G_K(\pi),$$

which cannot be captured by a single pairwise difference. To measure deviations from this condition, we define the *Qualification Gain Variance (QGV)*:

$$\text{QGV}(\pi) = \frac{1}{K} \sum_{k=1}^K \left(G_k(\pi) - \bar{G}(\pi)\right)^2, \quad \bar{G}(\pi) = \frac{1}{K} \sum_{k=1}^K G_k(\pi).$$

**Proposition 1.** *Let $G_k(\pi)$ and $\text{QGV}(\pi)$ be defined as above. Then:*

*1. $\text{QGV}(\pi) \geq 0$;*

*2. $\text{QGV}(\pi) = 0$ if and only if $G_1(\pi) = \cdots = G_K(\pi)$;*

*3.*

$$\text{QGV}(\pi) = \frac{1}{2K^2} \sum_{i=1}^K \sum_{j=1}^K (G_i(\pi) - G_j(\pi))^2;$$

*4. for $K = 2$,*

$$\text{QGV}(\pi) = \frac{1}{4} \left(G_1(\pi) - G_2(\pi)\right)^2.$$

*Proof.* Properties (1) and (2) follow directly from the definition of variance.

For (3), we expand:

$$\sum_{i=1}^{K}\sum_{j=1}^{K}(G_i - G_j)^2 = \sum_{i,j}(G_i^2 - 2G_iG_j + G_j^2)$$

$$= K\sum_{i=1}^{K}G_i^2 - 2\left(\sum_{i=1}^{K}G_i\right)^2 + K\sum_{i=1}^{K}G_i^2$$

$$= 2K\left(\sum_{i=1}^{K}G_i^2 - \frac{1}{K}\left(\sum_{i=1}^{K}G_i\right)^2\right).$$

On the other hand, by definition of variance,

$$\sum_{k=1}^{K}(G_k - \bar{G})^2 = \sum_{k=1}^{K}G_k^2 - \frac{1}{K}\left(\sum_{k=1}^{K}G_k\right)^2.$$

Combining the two expressions yields

$$\sum_{k=1}^{K}(G_k - \bar{G})^2 = \frac{1}{2K}\sum_{i=1}^{K}\sum_{j=1}^{K}(G_i - G_j)^2,$$

which proves (3).

For (4), when $K = 2$, we have $\bar{G} = (G_1 + G_2)/2$, and thus

$$\text{QGV} = \frac{1}{2}\left[\left(\frac{G_1 - G_2}{2}\right)^2 + \left(-\frac{G_1 - G_2}{2}\right)^2\right] = \frac{(G_1 - G_2)^2}{4}.$$

$\square$

### G.4 Interpretation

The above result shows that QGV is proportional to the average squared pairwise qualification-gain disparity. Hence, it constitutes a natural $K$-group generalization of the binary squared QGD penalty. In particular, minimizing QGV is equivalent to jointly minimizing disparities across all group pairs.

The equivalence also explains why optimization may become more challenging as the number of groups increases. QGV aggregates fairness pressure across all group pairs, which can dominate the utility objective if the fairness coefficient is not appropriately scaled.

