# OpenReview forum: "When Do Causal Fairness Constraints Work? Reproducing and Stress-Testing Long-Term Fair Reinforcement Learning"
_TMLR — Under review for TMLR_

### Review · Reviewer_CJFY · 2026-06-26

**Summary Of Contributions:**

The authors have taken the method and implementation of ““A Causal Lens for Learning Long-Term Fair Policies” ([pdf](zotero://open-pdf/library/items/SW5VVUY8?page=1)) , analyzed it, verified it, and extended it in mainly two  ways. The original work is a causal fairness analysis using RL, of a bank lending task where the lender tries to maximies profits while at the same time be fair with respect to a binary (K=2) sensitive attribute in the population. The oiriginal paper examined several RL learning algortihms, and anlayzed them with respect to qualification gain disparity (QGD). The original paper also decomposed the QGD into short-term and long-term fairness effects.

The authors find support in their reproduction with respect to the key key original findings, but nuance the discussion. Fairness, measured with QGD comes at a larger cost measured in profits, than the original publication portrayed.

The first extension to the original method is analysis of bank lending with K>2 groups with respect to the sensitive attribute. The fairness metric to measure fairness here is the qualification gain variance (QGV) which is a natural extension of  QGD.

The second extension of the method is the poposal of an infectous disease environment. Here, groups are defined structurally from the graph of nodes in a SIR model. Actions the agent takes is decisions about allocating limited vaccines to the different groups. The environment is different from the bank lending task in important ways, such as spillover effect from agent decisions on one groups vizavi other groups. Here, there are K>2 groups. They find that in this environment fairness has a very steep cost with respect to the utility.

**Additional Comments:**

Minor errors:

- section 2, missing a ‘dis’ in “…propose qualification gain parity (QGD) for a long-term fairness…”. this is somewhat critical since it introduces a load bearing term.

- page 10, Figure ??

- page 11, “Claim 4” is on wrong page with respect to the text it is a heading for. use \paragraph or \subsubsection or such, to make sure headings are not orphaned

- introduction of abbreviations are customaritly done with writing out the full name, and abbreviation in parenthesis. e.g. deep neural network (DNN). you have opted for a style of italization and reversal. *DNN (*deep neural network). while the choice is aestethical, i think your selected variant has more visual clutter and is harder on the eyes. Sometimes you also boldface to introduce terms instead of intlization. and sometimes, like in the case of SIR models, you did it the conventional way. this is inconsistent and gives the impression you did not proof read the submission.

**Audience:**

Yes

**Audience Explanation:**

The community is still very much grappling with accurate, grounded, and effective measures of fairness. this work shows that RL is a feasible way to discuss and analyze fairness in decision making, and that it can be useful in situations where simple causal analysis based on interventional units fail (e.g. network effects when intervening in a social network).

Reproduction papers are also very much needed, and I think we all benefit from more such work getting published and read. This format of reproduction + extensions makes it even more interesting and valuable.

**Broader Impact Concerns:**

I appreciate that the authors have included a discussion about compute used, and climate impact.

I would appreciate if the authors would briefly discuss the issues of operationalizing fairness as a quantitative metric. I know it is customary in the subfield of fairness metrics to take the assumption that fairness can be measured in quantitative terms based on outcomes, but this is in itself a ethical position to take. A single paragraph and a reference in the introduction might be enough, positioning this work within the broader ethical landscape (e.g. outcome fairness, procedural fairness etc)

**Claims And Evidence:**

No

**Claims Explanation:**

I think the presentation is too confusing for me to say it is ‘clear’ and ‘convincing’. I feel the authors probably have done all the analyses necessary, but fail to convey it. I lack a clear ‘contributions’ statement, that delineates what is pure reproduction, what is extended analysis, and what is additional analyses on top of the original paper. lacking this, there seems to sometimes be a slight mix up in the presentation what is results from the original paper, and what is contributions.

**Requested Changes:**

Critical changes

- Discuss the choice of using the original implementation. Reproductions relying on others code increase the risk of bugs in the orignial implementation being carried over. Did you do any vetting of the original implementation before using it? If not, why not?

- introduce an new section of subsection ‘contributions’ or similar that clearly delineates what your contributions are from the previous works. What is just reproduction and verification of original author claims? what is your added stress tests/verification analyses of the original method? what is the new method? what is the analysis and verification of the new method? and to be clear, with ‘method’ I mean both training objective, environments and metrics for analyses. this new section may replace the “Scope of Reproducibility” section.

- you say that hyperparameter/setting tuning is key. I would like to see some systematic investigation of this. and a discussion about plausibility for real world settings. can we sweep the parameter there?

- the infectious disease environment is explicitly only for a given graph (zavhary karate club, 34 nodes). I expect that all findings are heavily contingent on the graph structure. using random graphs and verifying finding robustness across different graphs seems critical to get any external validity of your findings. alternatively, if this is too much work at this stage, the discussion must make it clear that the results are of limited applicability, and discuss in what ways the selected graph was a reasonable representative choice


non-critical changes

- introduce the lending K>2 environment before the infectous disease environment. this will build up your contributions in a logical process starting in the original environment and progressively adding complexity.

- QGD/QGV requires one to define an underlying metric that the ‘gain’ is measured against. in the lending task this seems to be the credit score. it is not clear what it is for the infectous disease environment? vaccination status? health? healthy friends that one have contact with? make it clearer that the dyanmics of the environment is independent from the choice of variable with respect to which we measure fairness.

---

### Review · Reviewer_YiFD · 2026-07-07

**Summary Of Contributions:**

The paper „When do Causal Fairness Constraints Work? Reproducing and Stress-Testing Long-Term Fair Reinforcement Learning” presents the results of a replication study. The original study by Lear & Zhang introduced a group fairness objective that considers the whole trajectory of an agent in a RL setting (QGB) and extended PPO with penalty terms when this objective is violated (PPO-C). PPO-Cb extends this further to suppress that individuals are strong outliers from group fairness. The authors partically perform an exact reproduction of earlier results, confirming mostly the original results. They conceptually replicate the results on new data, finding strong deviations with respect to claims regarding utility preservation. They conceptually extend the evaluation to multi-group fairness showing the need for hyperparameter tuning to still preserve utility – though these experiments are not run on the new data.

**Audience:**

Yes

**Audience Explanation:**

Fairness is relevant and PPO an important RL algorithm.

**Claims And Evidence:**

No

**Claims Explanation:**

I strongly support replication studies and believe this is (mostly) well done and also highlighting why this is needed in science, since some results did not transfer to other settings. However, there are some strange choices which, to some extent, hamstring this work:

1) It is unclear to me, why group fairness considerations with k>2 is only evaluated on the financial data, not in the new disease setting. This work clearly shows that evaluating on a single data set can be misleading, shown by the strongly divergent results with respect to utility for the disease data. This greatly limits the soundness of the results here. Importantly, utility is already a problem with the new data. Whether the claims that group imbalance “does not come at the cost of degenerate utility” would be true in both settings is questionable, similarly, whether good results for the K-group setting can be achieved for the disease data. In other words, the new results are – empirically – as weak as the original results from an external validity perspective, which should not be the case in a replication study which – by definition – has the goal to *increase* the validity.

2) The results with respect to utility on the new data clearly and strongly contradict the original results. This is a crucial finding, because it puts in doubt whether using PPO-C is actually a good idea, as it may break utility. Unfortunately, there are no further insights here: neither properties of the data leading to this are analyzed (there is some mention of super-spreaders, but this is to neither specific, nor a generalizable property of data). There attempts to look beyond this for other data sets to further evaluate this discrepancy. The reporting of this is also fairly weak. Right now, there are two studies that basically look at “n=1” settings (i.e., the original one with one data set and this one with one data set) that are contradictory for – arguably – the most important RL property, i.e., utility. This means, right now, any claims regarding utility preservation are spurious. From how the results are written here, this is not clear.

Overall, I therefore have mixed feelings. On the one hand, the study revealed that the original results are clearly not fully sound. On the other hand, this new study is, from an empirical perspective, not really stronger with just one more project, meaning that we basically just learn that the results are not yet supported, but we do not learn really anything about when this works and when it does not, which would require a larger “n”.

Minor:

- Some references are not in brackets, mostly in Section 3.
- There is broken reference to a figure on page 10.
- $\pi_{PS}$ is not specified in Section 3.4
- There is no conclusion, which should be added.

**Requested Changes:**

EITHER justify why only a single new project is studied and why not all experiments were at least done in both settings. This needs to include why the results are sound, nevertheless.

OR extend the study with more experiments to provide a sound empirical basis.

---

### Review · Reviewer_SAvE · 2026-07-22

**Summary Of Contributions:**

The paper investigates the reproducibility and robustness of the method introduced in Lear & Zhang (2025) for tackling long-term fairness RL problems via causal constraints. The robustness of the proposed method was tested via additional benchmarks: a non-binary setting (K > 2) and the introduced metric to handle the setting, an imbalanced population setting, and a new problem domain (infectious disease) beyond the financial lending domain originally employed. The empirical investigation demonstrated that most of the results reported in the original paper are reproducible, while highlighting several conditions under which the method's performance is limited. In particular, the utility-fairness trade-off proved more fragile than originally reported, with utility preservation weaker than claimed in some lending settings, the K-group fairness penalty prone to collapsing utility when its coefficient is untuned, and a severe utility cost observed when the method was transferred to the infectious-disease domain.

**Additional Comments:**

N/A

**Audience:**

Yes

**Audience Explanation:**

The paper shed light on a relevant problem of long-term fairness in sequential decision-making agents, a growing research area that is relevant for the general ML community.

**Claims And Evidence:**

Yes

**Claims Explanation:**

The claims in the paper are supported by empirical evidence.

**Requested Changes:**

1. Section 2, line 1: change “qualification gain parity” to “qualification gain disparity”
2. Section 3.1.3, line 2: change parity to disparity
3. The line plot colour for PPO-C in Figure 6c is not clear from the graph. Please revise the plot.
4. Section 4.2, sub-title “Causal Decomposition of K-Group QGV”, paragraph 2, line 1: The figure number should be specified. It currently states: “Figure ??…”